# Polypharmacy in Older Adults with Alzheimer’s Disease

**DOI:** 10.3390/medicina58101445

**Published:** 2022-10-13

**Authors:** Satoru Esumi, Soichiro Ushio, Yoshito Zamami

**Affiliations:** 1Department of Pharmacy, Okayama University Hospital, 2-5-1 Shikata-cho, Kita-ku, Okayama 700-8558, Japan; 2The Faculty of Pharmaceutical Science, Kobe Gakuin University, 1-1-3 Minatojima, Chuo-ku, Kobe 650-8586, Japan

**Keywords:** Alzheimer’s disease, polypharmacy, drug interactions

## Abstract

The number of patients with Alzheimer’s disease is increasing annually. Most of these patients are older adults with comorbid physical illnesses, which means that they are often treated with a combination of medications for the disease they have and those for Alzheimer’s disease. Thus, older adults with Alzheimer’s disease are potentially at risk for polypharmacy. In addition, the drug interactions between Alzheimer’s disease medications and those for the treatment of physical illnesses may reduce their efficacy and increase side effects. This article reviews polypharmacy and drug interactions in elderly patients with Alzheimer’s disease, with a focus on psychotropic drugs.

## 1. Introduction

It is widely known that the number of older adults with dementia is increasing worldwide. In the United States, the number of people aged 65 years and older was 52 million in 2018 and is predicted to increase to 95 million by 2060, 50 million of which will have dementia by 2050 [1,2]. Similarly, the number of elderly with dementia in Asia is increasing. According to a meta-analysis published in 2014, the prevalence of dementia among patients aged 65 years and older in South Korea is 9.2%, which is higher than that of patients of similar age in Europe, the United States, and other Asian countries [3,4]. Furthermore, the odds ratio for death from dementia and other neurodegenerative diseases is reportedly higher in Korea than that in other countries. In Japan, there were an estimated 3.5 million people with dementia in 2012 (approximately 8% of the global population), and the number of dementia patients is projected to reach 4.9 million by 2034, when the population over 65 years of age reaches its peak [5]. In addition, the cost of informal care for patients with dementia is estimated at approximately $54 billion.

Most patients with Alzheimer’s disease are elderly and have two or more chronic conditions [6,7,8]. For example, a review of Medicare claims data revealed that 67% of beneficiaries over the age of 65 years had two or more chronic diseases [9]. Patients with Alzheimer’s disease and related dementias often have comorbid physical diseases [10]. Considering the barriers to communication and cognitive decline associated with this disease, polypharmacy is more likely to occur in patients with dementia than in the general elderly population [11,12]. According to the currently accepted definition, polypharmacy is defined as taking five or more medications per day, with a prevalence of 30–60% among the elderly (65 years and older) [13,14,15,16,17]. Although in most cases polypharmacy results from prescribing necessary medications to treat diseases in the elderly, there are reports that polypharmacy is associated with adverse outcomes. Inappropriate polypharmacy leads to an increased incidence of falls, frailty, and decreased quality of life [18,19,20,21]. Such adverse events further increase the cost and burden of care for dementia patients.

Patients with dementia have a high incidence of potentially inappropriate medication (PIM), estimated to be 14–74% [22,23,24]. PIMs are medications whose benefits do not exceed the risks associated with taking them, such as adverse events, and are a common cause of adverse drug reactions in the elderly [25,26]. They have been associated with a decreased quality of life, poor nutrition, and depression in nursing home residents with dementia [27,28,29]. According to a survey of PIM use among older adults with dementia in seven European countries, patients with dementia had a high incidence of PIM intake, regardless of severity, even those with mild Alzheimer’s disease [30,31]. Inadequate communication between patients with dementia and healthcare providers triggers a prescription cascade in which healthcare providers misidentify the side effects of drugs as new symptoms and prescribe drugs to treat drug-related problems, thereby triggering polypharmacy [32,33,34]. In other words, prescribing additional drugs for adverse reactions to PIM may lead to polypharmacy, which is associated with many more adverse events, greater healthcare utilisation, and even mortality [35,36].

Increased prescriptions are associated with a higher incidence of drug-drug interactions and adverse effects. Particularly in the elderly, drug interactions and a high sensitivity to psychotropic drugs may lead to unanticipated increases in effects and adverse events. In this study, we also review the drug interactions associated with cholinesterase inhibitors or memantine, both of which are used in the treatment of Alzheimer’s disease.

This review aimed to understand polypharmacy and drug interactions in patients with dementia (Figure 1). The contents of this review are summarised in Figure 2.

## 2. Polypharmacy in Alzheimer’s Disease Patients

The main symptoms of Alzheimer’s disease, namely memory impairment and behavioural and psychological symptoms, cause great distress to patients with dementia as well as caregivers [37,38]. Cognitive dysfunction in Alzheimer’s disease can be treated with cognitive-improving drugs, such as acetylcholinesterase inhibitors (donepezil) and NMDA-type glutamate receptor antagonists (memantine). Memantine and acetylcholinesterase inhibitors are sometimes used in combination to enhance efficacy; however, cognitive enhancers are generally used as a single medication [39]. Some countries do not have access to insurance coverage for cognitive-improving drugs because their effectiveness is limited [40].

The behavioural and psychological symptoms of dementia (BPSD) are customarily and widely treated symptomatically with psychotropic drugs [41,42]. Because of the wide variety of BPSD, including depression and delusions, the polypharmacy of psychotropic medication is increasing among the elderly, including patients without dementia [43]. The Beers Criteria, published by the American Geriatrics Society recommend avoiding use of more than three central nervous system (CNS)-active drugs, such as antidepressants, antipsychotics, benzodiazepines and “Z-drugs”, because they are associated with the risk of falls and fractures [44]. It is widely known that these drugs can produce falls due to sedation, daytime sleepiness, orthostatic hypotension, and motor disturbances.

Using Medicare claims data, a study examining psychotropic polypharmacy in patients with dementia revealed that 13.9% of older adults with dementia had “psychotropic polypharmacy” (three or more psychotropic medications for 30 or more consecutive days), and 29.4% of them were exposed to five or more psychotropic medications [1]. The drugs most frequently prescribed to patients with psychotropic polypharmacy were antidepressants (92.0% of days in polypharmacy during the study period), followed by antiepileptic drugs (62.1%), antipsychotics (47.1%), benzodiazepines (40.7%), opioids (32.3%), and benzodiazepine receptor agonists (6.0%). Although cognitive-improving drugs were not included in this study, considering that most dementia cases involve Alzheimer’s disease [45], it is assumed that the polypharmacy of CNS-acting drugs is even more serious with the added influence of cognitive-improving drugs.

A cross-sectional analysis using the National Ambulatory Medical Care Survey (NAMCS) showed that the number of medications was significantly higher in the elderly with dementia than in those without, although the number of outpatient visits did not differ based on dementia status [12]. In addition, the mean number of medications expected to be prescribed per visit was higher in patients with dementia when compared by age, sex, and number of comorbidities (standardised). Interestingly, elderly patients with dementia were more likely to be prescribed drugs that act on the CNS, as well as those that act on the gastrointestinal and urological systems and drugs for diabetes, indicating that polypharmacy with all drugs, not just CNS drugs for dementia symptoms, is likely to develop in patients with dementia.

Certain medications frequently included in polypharmacy, such as anticholinergics and sedatives, are associated with an increased risk of hospitalisation and death in patients with dementia [46]. Therefore, it is necessary to prescribe safer drugs and reduce the incidence of polypharmacy in this population [47,48]. The number of prescribed drugs should be reduced to a level consistent with the risk of hospitalisation and death in patients with dementia.

## 3. Cognitive Impairment Induced by Polypharmacy

Several reports have examined the association between polypharmacy and cognitive dysfunction, where polypharmacy is defined as the use of five or more drugs that are commonly used in clinical practice [9,17]. In a cross-sectional study on the association between polypharmacy and cognitive function and related comorbidities (depression, hypertension, and/or diabetes) in rural America, older adults afflicted with polypharmacy had 3.71 times higher odds of having cognitive impairment than older adults who were not [49]. In addition, even after adjusting for confounding factors using multivariate analysis, the odds of cognitive impairment were 2.86 times higher in patients with polypharmacy, whereas there was no significant association between comorbidities and cognitive impairment. Based on these results, it is considered that polypharmacy is independently associated with cognitive dysfunction. In general, patients with cognitive impairment have poor medication compliance. However, healthcare providers are unable to ascertain whether patients are taking their medications. Polypharmacy is thought to result from the addition of medications after determining that the prescribed medications are ineffective. Several reports have shown a negative correlation between the number of medications prescribed and adherence (i.e., the higher the number of medications, the worse the adherence) [50,51,52,53]. Future large cohort studies are needed to prospectively evaluate whether polypharmacy induces cognitive dysfunction or vice versa.

## 4. Drug Interactions in Alzheimer’s Disease

The risk of drug interaction is always present in patients undergoing polypharmacy. Since dementia patients with polypharmacy are prescribed not only psychotropic drugs but also multiple medications to treat physical illnesses, attention should also be paid to the status of physical illness medications. Since most patients with Alzheimer’s disease are elderly, the effects of aging must be considered in the pharmacokinetics of such patients. That is, muscle mass and total body water relatively decrease in the elderly, whereas body fat content increases [54,55,56]. These changes in body composition result in a reduction in the distribution volume of hydrophilic drugs and an increase in that of lipophilic drugs. In addition, drugs are more likely to reach the CNS in the elderly because of the reduced function of the blood-brain barrier [57]. The detoxification or disappearance of drugs absorbed into the body also declines in the elderly population. Hepatic blood flow decreases with age, and hepatic drug clearance can be reduced in the elderly [58,59]. Moreover, decreased renal blood flow is the most significant pharmacokinetic change associated with aging, leading to the decreased renal excretion of drugs.

### 4.1. Pharmacokinetic Drug Interactions

The major routes of disappearance and typical drug interactions of cognitive enhancers are listed in Table 1. Among the cognitive enhancers, donepezil and galantamine are metabolised in the liver via CYP2D6 and CYP3A4, and hepatic metabolism may be affected by specific substrates, inhibitors, or enhancers of the same enzymes [60]. The modes of metabolic inhibition by drug interactions fall into two major categories [61]: (1) direct enzyme inhibition (e.g., ketoconazole, a strong non-competitive inhibitor of CYP3A4), and (2) competitive inhibition of the catalytic site of the CYP3A4 enzyme. Ketoconazole significantly increased the plasma concentration of donepezil, presumably due to the inhibition of CYP3A4 [62]. In addition, according to the U.S. FDA, many drugs, such as ritonavir and other antivirals, clarithromycin and other macrolide antibiotics, and verapamil and fluvoxamine, have CYP3A4 inhibitory effects. The effect of CYP2D6 on donepezil metabolism is unclear, and donepezil is assumed to be less susceptible to renal or hepatic impairment in patients. Galantamine metabolism has been reported to be affected by both CYP2D6 and CYP3A4 inhibitors. The combination of galantamine with ketoconazole (a strong CYP3A4 inhibitor) or paroxetine (a strong CYP2D6 inhibitor) increased the area under the blood concentration-time curve (AUC) for galantamine by 30% and 40%, respectively, compared with the administration of galantamine alone [63]. Several antidepressants (paroxetine, fluoxetine, bupropion, duloxetine, escitalopram, fluvoxamine, and sertraline) have been alerted by the FDA to have CYP2D6 inhibitory activity to varying degrees. It was also noted that galantamine increases blood levels by approximately 30–60% in patients with impaired renal or hepatic function. While donepezil is not shown to exacerbate renal or hepatic impairment, blood levels of donepezil is increased in patients with renal or hepatic impairment. Therefore, greater caution should be exercised when galantamine is combined with other drugs in elderly patients with impaired renal or hepatic function, because both physiological changes and drug interactions can increase blood concentrations. Regarding drug classes, among psychotropic drugs, antidepressants should be used with caution because they may affect the pharmacokinetics of donepezil and galantamine.

Rivastigmine is rapidly metabolised by esterases in the blood, and is not metabolised by cytochrome P450. Thus, there was no risk of cytochrome P450-related drug interactions, and no significant differences in the ability to metabolise rivastigmine between patients with Alzheimer’s disease and healthy older adults were observed [64]. It was reported that rivastigmine does not increase adverse events in patients with Alzheimer’s disease when used in combination with 22 different drugs, including antidiabetic, cardiovascular, gastrointestinal, and nonsteroidal anti-inflammatory drugs [65].

Memantine is a weak base drug with a pKa of 10.27 and is excreted mainly from the kidneys as an unchanged drug [66,67]. Urinary pH has been shown to significantly affect the disappearance of drugs excreted renally in an unchanged form, such as memantine, and the acidification of urine increases excretion and decreases blood concentration [68]. Conversely, the marked alkalinisation of urine pH induces a decrease in memantine excretion, which may lead to the overexposure of tissues to memantine and toxic effects, especially in elderly patients with reduced renal function. Therefore, the use of drugs or foods that alkalize urine, such as acetazolamide, citric acid, and sodium bicarbonate, may decrease the excretion of memantine. In addition, renal cation transporters, particularly multidrug and toxin extrusion proteins (MATE1), may be involved in memantine excretion [69].

### 4.2. Pharmacodynamic Drug Interactions

Typical pharmacodynamic interactions of cognitive enhancers are summarized in Table 2. Because acetylcholine neurotransmission in the brain is reduced in patients with Alzheimer’s disease [70], a cognitive enhancer that increases acetylcholine in synaptic terminals by inhibiting cholinesterase in the brain has been developed. Anticholinergics have opposite mechanisms of action relative to cholinesterase inhibitors, thus their concomitant use may attenuate their mutual action [71].

Clinically used psychotropic drugs, including many tricyclic antidepressants (e.g., amitriptyline, amoxapine, and imipramine) and other antidepressants (nortriptyline and paroxetine), many first-generation antihistamines (e.g., diphenhydramine and hydroxy persons), anticholinergic antiparkinsonian agents (trihexyphenidyl and biperiden), and several antipsychotics (chlorpromazine, perphenazine, olanzapine and clozapine) [44]. As already noted, these drugs and cognitive enhancers are frequently administered to patients with Alzheimer’s disease. In fact, the rate of concomitant anticholinergic medications in patients using donepezil was found to be higher than that of elderly patients in the control group [44]. In addition, an increased prescription of anticholinergics at the start of cholinesterase inhibitor therapy, which is often inappropriately used, has also been reported [72]. One reason for this combination may be that atropine and other anticholinergics are used in combination with cholinesterase inhibitors to reduce their adverse effects [73]. Dose optimisation and minimisation of adverse events due to titration may be the most important factors in the use of these drugs for patients receiving cholinesterase inhibitor therapy [74].

Memantine is known to act on NMDA-type glutamate receptors. Therefore, the concomitant use of memantine with drugs that act on NMDA-type glutamate receptors (amantadine, ketamine, and dextromethorphan) may result in the competitive inhibition of its action [75]. Although reported at a nonclinical level, memantine’s NMDA receptor antagonism enhances dopamine release in the prefrontal cortex and striatum [76]. Decreased dopamine levels in the prefrontal cortex are known to be involved in cognitive-learning impairment and negative symptoms [77,78]. In addition, many antipsychotic drugs block dopamine receptors in these brain regions, as well as in the nucleus accumbens, resulting in negative symptoms and cognitive dysfunction. The efficacy of memantine in the treatment of learning and cognitive dysfunction and the negative symptoms of schizophrenia has become increasingly clear in recent years, although its effects on hallucinations via dopamine neurotransmission in the nucleus accumbens should be carefully monitored [79,80].

## 5. Conclusions

This study outlines the current state of the polypharmacy of drugs used in patients with Alzheimer’s disease and the drug interactions with cognitive enhancers. Patients with Alzheimer’s disease have increased sensitivity to psychotropic drugs owing to their decreased brain function and delayed drug elimination due to aging. These factors make them susceptible to polypharmacy and drug interactions. Adverse drug events can decrease the patients’ quality of life and worsen their prognosis. Considerable effort should be made to improve patients’ quality of life.

## Figures and Tables

**Figure 1 medicina-58-01445-f001:**
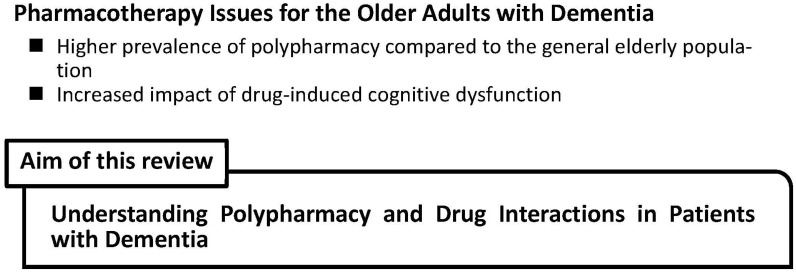
Aim of this review.

**Figure 2 medicina-58-01445-f002:**
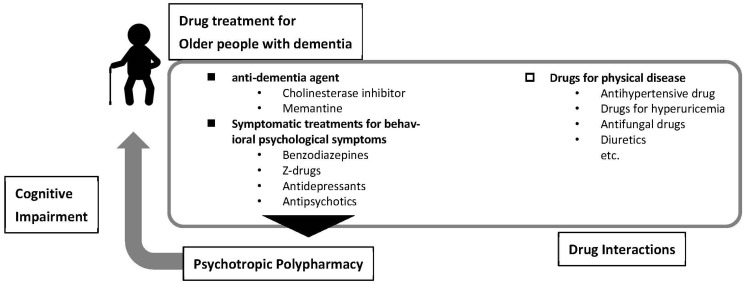
Summary of contents in this review.

**Table 1 medicina-58-01445-t001:** Pharmacokinetic Drug Interactions in Alzheimer’s Disease.

Drugs	Extinction Pathway	Mechanisms of Drug Interactions	Typical Drugs in Drug Interactions
Donepezil Galantamine	Hepatic metabolism (CYP3A4 and CYP2D6)	Inhibition of CYP3A4	ketoconazole, itraconazole, erythromycin, ritonavir, atazanavir
		Inhibition of CYP2D6 (especially, in galantamine)	bupropion, fluoxetine, paroxetine, quinidine
		Induction of CYP3A4	carbamazepine, phenytoin, rifampin
Rivastigmine	Metabolism by blood esterase	not reported	
Memantine	Urinary excretion	Change in urinary pH	acetazolamide, citric acid, sodium bicarbonate

**Table 2 medicina-58-01445-t002:** Pharmacodynamic Drug Interaction in Alzheimer’s Disease.

Drugs	Mechanisms of Drug Interactions	Typical Drugs in Drug Interactions
Donepezil Galantamine Rivastigmine	Attenuation of antidementia drugs by anticholinergic action	trihexyphenidyl, biperiden, butylscopolamine, atropine amitriptyline, clomipramine
	Enhanced peripheral cholinergic stimulation by cholinergic agonist	acetylcholine, betanechol, distigmine, neostigmine
Memantine	Enhancement of NMDA receptor antagonism	amantadine, ketamine, dextromethorphan

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
