# Peer review of "Polypharmacy in Older Adults with Alzheimer’s Disease"

_medicina, 2022, doi:10.3390/medicina58101445_

Round 1

Reviewer 1 Report

The manuscript entitled Polypharmacy in Older Adults with Alzheimer’s Disease is novel  manuscript. The following comments should be considered before acceptance.

1. An english editing of the manuscript should be performed

2. Two figures at list should be included of the review article describing the main aim of the review and the content

3. Two tables at least should be included in the manuscript, for example the different types of interactions. .....

Author Response

Reviewer 1

Thank you for your comments that were helpful in improving the content.

We have revised the text according to your comments. The revisions are highlighted using the change log record. In addition, duplicate citations were found and have been corrected.

  1. An english editing of the manuscript should be performed

Response:

We have already commissioned Editage to perform the English editing. We have changed the text according to the reviewer's comments, therefore we have performed English editing again.

  1. Two figures at list should be included of the review article describing the main aim of the review and the content

Response:

In accordance with the reviewer's comments, we prepared two figures on the aim and content of our review.

  1. Two tables at least should be included in the manuscript, for example the different types of interactions.

Response:

In accordance with the reviewer's comments, we have created two tables summarizing the pharmacokinetic and pharmacodynamic interactions.

Reviewer 2 Report

The paper "Polypharmacy in Older Adults with Alzheimer's Disease" aims to review the polypharmacy and drug-drug interactions in old patients with Alzheimer's disease with a focus on psychotropic drugs.

The paper is well documented, as reflected by the number of references, is well structured, but still misses—in my opinion—some major points:

  1. In the Introduction section, lines 56-59 must be revised by the authors; it seems to be a repetition of the same idea under two slightly different formulations. Same issue on lines 126-129.
  2. Lines 78-80 are confusing: all types of polypharmacy are associated with the risk of falls and fractures? The authors must add certain (classes of) drug which correlates with falls and fractures; otherwise, the readership can understand that a combination of lowering blood pressure drugs, diuretics, and lipid-lowering drugs could lead to falls and fractures, which is not correct. Hence, please insert here some examples, such as benzodiazepines, with brief explanations of the mechanism leading to falls and fractures. Make these lines sound like it deals with psychotropic polypharmacy.
  3. Lines 108-111: what are the main conclusions of references 51 and 52? Does polypharmacy correlate with cognitive impairment? Do some specific classes of drugs correlate with cognitive impairment? Are some specific comorbidities that involve polypharmacy and correlate with cognitive impairment?
  4. Line 119 introduces a new paragraph that presents "these results"; which results? of the first paragraph?
  5. In section 4.1, Pharmacokinetic drug interactions, the authors should bring for memantine some examples of drugs that influence the urinary pH and consequently the excretion of memantine. 
  6. As the abstract and conclusion underline that the main topic is drug interactions in elderly patients with Alzheimer's disease with the focus on psychotropic drugs, the authors should come up with more representative examples of psychotropics in each section and subsection, let alone they need to clarify where psychotropics are in combination with drugs for Alzheimer's disease.

Minor issues:

  1. Line 55—I suggest changing the verb "trigger" with something more appropriate for "polypharmacy," such as "contributing to"
  2. Line 90: keep consistency in indexing the reference (see reference 46)
  3. In the second section, line 76, the expression "palliative psychotropic drugs" does not read well and is incorrect from a pharmacologic standpoint; perhaps "psychotropic drugs used in palliative care" would be better. 

Author Response

Reviewer 2

Thank you for your careful peer review and accurate comments.

We have revised the text according to your comments. The revisions are highlighted using the change log record. In addition, duplicate citations were found and have been corrected.

  1. In the Introduction section, lines 56-59 must be revised by the authors; it seems to be a repetition of the same idea under two slightly different formulations. Same issue on lines 126-129.

Response:

As the reviewer directed, we removed the similar content.

  1. Lines 78-80 are confusing: all types of polypharmacy are associated with the risk of falls and fractures? The authors must add certain (classes of) drug which correlates with falls and fractures; otherwise, the readership can understand that a combination of lowering blood pressure drugs, diuretics, and lipid-lowering drugs could lead to falls and fractures, which is not correct. Hence, please insert here some examples, such as benzodiazepines, with brief explanations of the mechanism leading to falls and fractures. Make these lines sound like it deals with psychotropic polypharmacy.

Response:

Thank you for your important comments. The description has been changed to one that deals with multiple doses of psychotropic drugs as you pointed out.

  1. Lines 108-111: what are the main conclusions of references 51 and 52? Does polypharmacy correlate with cognitive impairment? Do some specific classes of drugs correlate with cognitive impairment? Are some specific comorbidities that involve polypharmacy and correlate with cognitive impairment?

Response:

Thank you for your important comments. The text sought to indicate that there are a small number of reports showing an association between polypharmacy and cognitive dysfunction, but what was stated was the exact opposite of what was intended. We have corrected the text based on your remarks.

  1. Line 119 introduces a new paragraph that presents "these results"; which results? of the first paragraph?

Response:

“These results” refers to the results in the previous paragraph, that older adults with polypharmacy conditions have higher odds ratios for cognitive impairment. The paragraphs have been merged accordingly.

  1. In section 4.1, Pharmacokinetic drug interactions, the authors should bring for memantine some examples of drugs that influence the urinary pH and consequently the excretion of memantine.

Response:

In accordance with the reviewer’s comments, examples of drugs that affect urinary pH and thus affect memantine excretion are listed. The effects of these drugs on urinary pH are widely known but could not be found in the literature, so citations could not be provided.

  1. As the abstract and conclusion underline that the main topic is drug interactions in elderly patients with Alzheimer's disease with the focus on psychotropic drugs, the authors should come up with more representative examples of psychotropics in each section and subsection, let alone they need to clarify where psychotropics are in combination with drugs for Alzheimer's disease.

Response:

In accordance with the reviewers' comments, the number of representative examples of psychotropic drugs to be discussed in each article was increased.

Minor issues:

  1. Line 55—I suggest changing the verb "trigger" with something more appropriate for "polypharmacy," such as "contributing to"

Response:

As the reviewer indicated, we corrected the verb.

  1. Line 90: keep consistency in indexing the reference (see reference 46)

Response:

As the reviewer indicated, we corrected citation formatting.

  1. In the second section, line 76, the expression "palliative psychotropic drugs" does not read well and is incorrect from a pharmacologic standpoint; perhaps "psychotropic drugs used in palliative care" would be better.

Response:

As the reviewer indicated, we corrected the sentence in line 76, as follows “The behavioural and psychological symptoms of dementia (BPSD) are customarily and widely treated symptomatically with psychotropic drugs.”

Reviewer 3 Report

A good paper, publishable as it is.

Author Response

Dear reviewer #3,

Thank you for your appropriate peer review and favorable comments.

Round 2

Reviewer 1 Report

No comments

Reviewer 2 Report

The authors responded to most of the issues raised in the previous review. However, their modifications are challenging to follow in the revised manuscript, as the lines have gaps from the initial manuscript. I recommend that the authors indicate the lines where the changes appear in the revised manuscript to each reviewer.

Although the authors did not address the minor issue of using the verb "triggers," they claim they did. 

Line 92I recommend that authors explain what Z-drugs are.

Lines 228-232The phrase has no verbal predicate; what is its message?